# One-Step Solid-State Synthesis of Ni-Rich Cathode Materials for Lithium-Ion Batteries

**DOI:** 10.3390/ma16083079

**Published:** 2023-04-13

**Authors:** Lifan Wang, Qinling Shi, Chun Zhan, Guicheng Liu

**Affiliations:** 1State Key Laboratory of Advanced Metallurgy, School of Metallurgical and Ecological Engineering, University of Science and Technology Beijing, Beijing 100083, China; 2Department of Energy Storage Science and Engineering, School of Metallurgical and Ecological Engineering, University of Science and Technology Beijing, Beijing 100083, China; 3School of Energy Power and Mechanical Engineering, North China Electric Power University, Beijing 102206, China; 4Department of Physics, Dongguk University, Seoul 04620, Republic of Korea

**Keywords:** lithium-ion batteries, Ni-rich cathode materials, one-step solid-state synthesis, electrochemical performance

## Abstract

Ni-rich cathodes are expected to serve as critical materials for high-energy lithium-ion batteries. Increasing the Ni content can effectively improve the energy density but usually leads to more complex synthesis conditions, thus limiting its development. In this work, a simple one-step solid-state process for synthesizing Ni-rich ternary cathode materials NCA (LiNi_0.9_Co_0.05_Al_0.05_O_2_) was presented, and the synthesis conditions were systematically studied. It was found that the synthesis conditions have a substantial impact on electrochemical performance. Furthermore, the cathode materials produced through a one-step solid-state process exhibited excellent cycling stability, maintaining 97.2% of their capacity after 100 cycles at a rate of 1 C. The results show that a one-step solid-state method can successfully synthesize Ni-rich ternary cathode material, which has great potential for application. Optimizing the synthesis conditions also provides valuable ideas for the commercial synthesis of Ni-rich cathode materials.

## 1. Introduction

As highly efficient electrochemical energy storage devices, lithium-ion batteries have broad application prospects and commercial value. With cruising duration being particularly important in electric vehicles (EVs), increasing the specific capacity is of great significance [1,2]. The capacity of commercial anode materials is much larger than that of cathode materials, so the capacities of the cathode materials become a limiting factor, and studies regarding cathode materials with high specific energy are highly significant [3,4]. Presently, as the cathode material for high-energy lithium-ion batteries with the most potential, a Ni-rich layered structure (LiNi_x_Co_1−x_O_2_) has drawn much attention due to its low cost [5,6]. Unlike typical cobalt-based layered oxides, the Ni-rich layered oxides ensure the near-theoretical specific capacity and simultaneously lower the toxic Co. Furthermore, increasing Ni concentration enhances the specific capacity. For instance, ultra-high nickel oxides (LiNi_x_Co_1−x_O_2_ (x ≥ 0.9)) have been widely investigated [7,8,9,10]. Among them, LiNi_0.9_Co_0.05_Al_0.05_O_2_ (NCA) was developed for its high energy density and low price. These things considered, Al-doping has been proven to increase thermal stability and safety, as well as improve the cycling performance of the electrodes [11,12]. Additionally, Al-doping has been proven to stabilize the charge-transfer impedance on the cathode, leading to a successful reduction in cell impedance [13]. Therefore, Al-doping is essential to lowering the cost and improving the stability of NCA, thereby reducing the cost of lithium-ion batteries [14].

The structures and synthesis conditions of cathode materials will affect their cost and electrochemical performance. However, with a higher Ni^3+^ content in Ni-rich layered oxides, the synthesis conditions are more stringent and require precise control of calcination temperature, mixing time, and calcination atmosphere [15,16,17,18]. For the synthesis process of Ni-rich layered materials, the main synthesis method includes the sol–gel method, co-precipitated method, hydrothermal method, and solid-state method. The sol–gel method uniformly mixes the precursors in the liquid phase and carries out hydrolysis and condensation chemical reactions, forming a stable sol-system, which coalesces slowly and gradually forms a gel with three-dimensional space network structures [19]. The combining and refinement of raw ingredients using the sol–gel method offers clear advantages, but the preparation required is difficult. The co-precipitated method is widely used for the current commercial synthesis of ternary cathode materials by stirring and mixing the stoichiometric ratio of transition metal salt solutions to form homogeneous precursors [20,21]. This process requires the precise control of parameters such as stirring rate, pH, and initial concentration [22,23,24,25]. After thorough mixing with the lithium source and sintering at high temperatures, the cathode materials with uniform particle shapes and sizes are synthesized. Even though this process is relatively mature, it is costly, complicated, and limited by the elements. It is difficult to prepare precursors containing aluminum using the co-precipitated method because Al^3+^ ions have a propensity to form double hydroxides in solution [26]. Therefore, notable alternatives to co-precipitation for the synthesis of NCA are presented. The hydrothermal method involves dissolving the metal compound in an aqueous solution and then generating the cathode material in a closed vessel after a high temperature and pressure reaction, but the operation is complicated. The solid-state method directly mixes lithium sources with transition metal salts via ball milling. The powder is then pre-sintered at a low temperature and calcined at a high temperature after grinding to obtain cathode materials. The advantages of this method include simple operation and low cost, but it is hard to mix homogeneously [27,28]. Zheng et al. successfully synthesized single-crystal Ni-rich cathode materials with a layered structure via a two-step solid-state method [29]. However, this method has only two steps and is still tedious.

The above discussions raise questions about whether these methods most effectively aid the synthesis of Ni-rich oxides LiNi_0.9_Co_0.05_Al_0.05_O_2_ or whether a simple and costless process is reasonably necessary. In this work, we successfully develop a simple one-step solid-state method to synthesize NCA. Our results show that the synthesis process costs less and is more straightforward than the conventional process. Additionally, the NCA material has comparable electrochemical performance with the materials synthesized by commercial co-precipitated methods. Moreover, the uniform distribution of Al enhances the stability by inhibiting the irreversible phase transition, also confirming the homogeneity of the material. In the further adjustment of synthesis conditions, we found that the electrochemical performance is greatly influenced by the grain size and degree of grain breakage. Therefore, by modifying these synthesis conditions, we can synthesize NCA through this one-step solid-state method with enhanced performance. This work explores the effect of Al-doping by comparing the performance and structure of ternary materials LiNi_0.9_Co_0.05_Al_0.05_O_2_ and binary materials LiNi_0.9_Co_0.10_O_2_, indicating the feasibility of the one-step solid-state method. Furthermore, the optimal conditions and their mechanism for performance enhancement are established by carefully controlling the calcination temperature, the spherical material ratio, and different precursors.

## 2. Experimental Section

### 2.1. Synthesis Procedures

LiNi_0.9_Co_0.05_Al_0.05_O_2_ materials were synthesized via a one-step solid-state method. We combined LiOH∙H_2_O (AR, Aladdin), Ni(OH)_2_ (AR, Aladdin), Co_3_O_4_ (99.9%, Aladdin), and Al_2_O_3_ (99.9%, Aladdin) in stoichiometric proportions (Li:TM = 1.05:1 and Ni:Co: Al = 90:5:5) and ball grinding beads were put in a ball mill jar with anhydrous ethanol, which is appropriate for this process. The mixture was ground at 300 r/min for 6 h and subsequently poured into a beaker before being completely dried in a blast oven at 80 °C. After drying, the powder was manually mixed via hand grinding to achieve a homogeneous consistency. The powder mixture was sintered at 750 °C for 12 h under oxygen flow in a tube furnace and cooled to room temperature. The final sample is referred to by the weight ratio of the ball to material–aluminum precursor types and calcination temperatures. For example, the sample with the ratio of ball to powder at 7 with Al_2_O_3_ precursors at 700 °C is called 7-Al_2_O_3_-700 °C.

### 2.2. Electrochemical Measurements

The detailed steps for the 2025 coin-type cell assembly are shown in Appendix A. Subsequently, the assembled 2025 coin-type cell is clipped to a special fixture in the Neware battery test system (CT-4008T, Neware, Shenzhen, China) to test the electrochemical performance. All the batteries were tested at a rate of 0.1 C (1 C = 180 mA/g) in the range of 2.8–4.3 V and 2.8–4.5 V vs. Li+/Li for 3 cycles and then tested at a rate of 0.5 C for 100 cycles. The galvanostatic intermittent titration technique (GITT) test was also performed on a Neware battery test system at 0.1 C with a pulse time of 10 min and a relaxation time of 60 min.

### 2.3. Material Characterization

To observe the morphology and the elemental distribution of the samples, we employed a Zeiss SUPRATM55 FESEM and an energy spectrometer (EDS). In order to study the crystal structure, the phase composition and lattice parameters of samples were determined by powder X-ray diffraction (XRD, D/Max 2000, Rigaku), equipped with a Cu Kɑ diffracted monochromator. Diffraction patterns were collected in the scattering angle (2θ) range of 10–90° with a scanning speed of 5°/min at 0.02° intervals. The test results were refined using Fullprof software 2.0. The cycled and pristine materials were characterized with X-ray photoelectron spectroscopy (XPS) using Shimadzu Kratos Axis Ultra DLD with an Al Kɑ diffracted monochromator.

## 3. Results and Discussion

### 3.1. Impact of Synthesis Conditions on the Electrochemical Performance

#### 3.1.1. Impact of Aluminum Precursors

The one-step solid-step method of synthesizing NCA materials is based on thoroughly mixing raw materials. Thus, the molecular mass and structure of different types of precursors may significantly affect the electrochemical performance during the synthesis process. In this work, three different aluminum precursors, Al_2_O_3_ (99.99%, Aladdin), Al(NO_3_)_3_ (AR, Aladdin), and Al(OH)_3_ (AR, Macklin), were used to synthesize the NCA materials at 750 °C with a ball/material ratio of 13:1. To fully investigate the effect of different aluminum precursors, the constant—current charge/discharge test was performed. As indicated in Figure 1a, the initial charge-specific capacities of the cathodes are 238 mAh/g (Al(NO_3_)_3_), 238 mAh/g (Al_2_O_3_), and 237 mAh/g (Al(OH)_3_), corresponding to discharge-specific capacities of 201 mAh/g, 206 mAh/g, and 200 mAh/g, with coulombic efficiencies of 84.4%, 86.5%, and 84.3%, respectively. The detailed data are given in Appendix A. In Figure 1b, the discharge capacity of NCA cathode materials synthesized utilizing different aluminum precursors (Al(NO_3_)_3_, Al_2_O_3_, and Al(OH)_3_) is shown. After 100 cycles at 1C, the discharge capacity of NCA cathode materials synthesized from Al(NO_3_)_3_ and Al_2_O_3_ precursors decays from 178 mAh/g to 157 mAh/g and 185 mAh/g to 163 mAh/g, respectively, with a capacity retention rate of 88.2% and 88.1%, while NCA material synthesized by Al(OH)_3_ decays from 178 mAh/g to 171 mAh/g with a capacity retention rate of 96.0%. Above all, the cathode material with Al(OH)_3_ as the precursor exhibited the highest capacity retention after 100 cycles. This could be attributed to the fact that Al(OH)_3_ and Ni(OH)_2_ are hydroxides, thus reducing the impact during mixing to a certain degree. To further study the differences, the rate performances are shown in Figure 1c, where no significant discharge capacity differences between the three materials are discernible until the rate is increased to 10 C, which is 155 mAh/g (Al(NO_3_)_3_), 165 mAh/g (Al_2_O_3_), and 160 mAh/g (Al(OH)_3_). The detailed data are listed in Appendix A. The dQ/dV curves are analyzed during the initial charging and discharging process (Figure 1d). All three materials have three pairs of redox peaks, but compared to other materials, the cathode material with Al(OH)_3_ precursor has a more pronounced phase transition peak at H1-M, which may be due to the higher crystallinity of the cathode material with Al(OH)_3_. The cathode material with Al_2_O_3_ exhibits a higher discharge capacity and better rate performance due to its most minor powder and the most uniform distribution. It is consistent with the electrochemical performance in the voltage range of 2.8–4.5 V (Appendix A). Additionally, because Ni(OH)_2_ is the nickel precursor, Al(NO_3_)_3_ is acidic during the mixing process, thus affecting the structure of the bulk phase and further resulting in poor electrochemical performance of the materials. Therefore, using Al(OH)_3_ and Al_2_O_3_ as aluminum precursors for Ni-rich materials results in improved performance.

#### 3.1.2. Impact of Ball Material Ratio

As a critical step in the one-step solid-state synthesis process, the optimization of high-energy ball mixing is significant, such as the ball/material ratio, mixing speed, and ball milling time. Among these parameters, the ball ratio plays a key role in determining the particle size and morphology of final products, which in turn can influence their electrochemical performance. In this work, we investigate the effect of three different ball/material ratios (1:7, 1:13, and 1:20) on the morphology and electrochemical performance of NCA materials synthesized using Al_2_O_3_ as the precursor at 750 °C. SEM tests were conducted on the material particles synthesized under different spherical material ratios. As shown in Figure 2, the size of the primary particles of the samples prepared with different spherical material ratios remains almost unchanged. By comparison, the micron spherical size of the primary particles agglomerated into secondary particles becomes larger as the sphere ratio decreases. The results demonstrate that, as the ball material ratio decreases, the particle size of the secondary particles becomes larger, which can lead to a larger surface area, minor surface resistance, larger surface energy, and slower capacity decay. However, larger secondary particle sizes create longer lithium-ion transport distances, resulting in a reduced capacity. It can be concluded that at smaller ball ratios of 1:7, the particles may not mix properly, which may result in a lower initial discharge capacity of the material, while at larger ball ratios of 1:20, it may lead to particle fragmentation, which in turn lowers the capacity retention during cycling.

Figure 2g shows the charging and discharging curves of NCA materials synthesized with the ball material ratios of 1:7, 1:13, and 1:20. The electrodes exhibit initial specific charge capacities of 225 mAh/g, 239 mAh/g, and 233 mAh/g, the corresponding discharging specific capacities are 189 mAh/g, 206 mAh/g, and 198 mAh/g, with coulombic efficiencies of 84.0%, 86.2%, and 84.9%, respectively. The detailed data are listed in Appendix A. The charge/discharge capacity increases and subsequently drops as the ball ratio rises, perhaps because a smaller ball ratio causes metal ions to mix less evenly, and a higher ball ratio can, to some extent, make metal ions mixed more consistently. Moreover, raising the ball ratio can make metal ions mix more consistently to a certain extent. However, when the ball ratio is too high, the grains of the precursor will be broken, which decreases the secondary particle size, and the grains cannot grow in time. The cyclic stability is tested after 100 cycles at 1 C in Figure 2h. The capacity retention rates of the materials synthesized with ball ratios of 1:7, 1:13, and 1:20 are 97.5%, 87.5%, and 81.9%, respectively. The capacity retention rate decreases with the increase of ball material ratios, which may be due to the particle fragmentation caused by excessive ball material ratio, which in turn affects the structural stability. In Figure 2i, the discharge capacity difference is noticeable when the rate turns to 10 C. The detailed data are listed in Appendix A. By comparing the dQ/dV curves during the initial cycling in Figure 2k, the material synthesized with a 1:13 ball material ratio entered the first electrochemical plateau before the other two ball material ratios, followed by 1:20, and finally 1:7, and the remaining two redox plateaus were found to be relatively overlapping. In the three conditions of this experiment, the materials with a 1:7 ball material ratio had a better cycling performance, and those with a 1:13 ball material ratio had a better rate performance than the others. This is consistent with the electrochemical performance in the voltage range of 2.8–4.5 V (Appendix A). Therefore, the adjustment of the spherical material ratio in the solid-state method is very effective in influencing the performance of the synthesized materials.

#### 3.1.3. Impact of Calcination Temperature

Figure 3 shows the particle size distribution patterns of NCA with a ball/material ratio of 13:1 and the Al_2_O_3_ aluminum precursor made at 700 °C, 725 °C, 750 °C, 775 °C, and 800 °C, respectively. As the temperature rises, the particle size gradually increases. When the temperature increases, the grain growth is adequate, and the grain size affects the Li^+^ transport rate, which in turn affects the electrochemical properties of the material. A low temperature is not conducive to grain growth and/or the adequate mixing of metal atoms, while a high temperature will easily result in abnormal grain growth. Electrochemical tests were performed on all the samples to explore the effect of this factor.

The effect of calcination temperatures on the electrochemical performance of the synthesized NCA materials was investigated through various electrochemical tests, as shown in Figure 4. As depicted in Figure 4a, the initial capacity of the material rises and then falls as the temperature increases. Additionally, when the temperature exceeds 750 °C, the discharge-specific capacity declines rapidly. This may be because the high temperature leads to the forming of oxygen vacancies, causing lattice defects and an increased resistance to lithium-ion insertion and extraction, eventually affecting the capacity. As shown in Figure 4b, after 100 cycles, the corresponding discharge capacity varies greatly with the temperatures. The capacity retentions of the synthesized NCA materials at 700 °C and 725 °C are 97.1% and 92.8%, respectively. The NCA material at 700 °C possesses the best cycling stability, with the capacity decaying from 177 mAh/g to 172 mAh/g. The detailed data are listed in Appendix A. Therefore, as the temperature reaches a specific limit, increasing the temperature will cause cation mixing and obstruct the channels for lithium-ion transport, thus decreasing the cycling stability. The NCA material synthesized at 725 °C exhibits the best rate performance, as depicted in Figure 4c, indicating that moderate grain growth is favorable for rate performance. The detailed data are provided in Appendix A. Figure 4d displays the dQ/dV curves of the materials synthesized at different temperatures. The materials at 775 °C have delayed access to the electrochemical plateau, and the intensity of the H2–H3 peak at 800 °C is reduced, finally affecting the discharge-specific capacity. This is consistent with the electrochemical performance in the voltage range of 2.8–4.5 V (Appendix A). This conclusion is also attributable to the high calcination temperature, which converts Ni^3+^ to Ni^2+^. Moreover, increasing the proportion of Ni^2+^ leads to an increased mixing degree of the materials.

### 3.2. Phase Composition and Surface Morphology of As-Prepared Cathode Materials

To fully investigate the phase structure, XRD analysis was conducted on LiNi_0.9_Co_0.05_Al_0.05_O_2_ (NCA) and LiNi_0.9_Co_0.05_O_2_ (NC) cathode materials. The ternary layered cathodes can be regarded as adding Al during the synthesis of binary nickel-cobalt materials, so they have similar structures. Appendix A shows the XRD patterns of NCA and NC materials. All diffraction peaks match the hexagonal α-NaFeO_2_ structure with space group R$\bar{3}$m, indicating the formation of a highly ordered layered structure. The characteristic doublets of (006)/(102) and (108)/(110) clearly show a significant splitting, suggesting the sample possesses a well-ordered layered structure. Moreover, the detailed refinement structural parameters in Appendix A show that the lattice parameters c/a ratio is 4.945 of NCA, greater than 4.8 of NC, indicating that the NCA materials possess an ordered layered structure. Additionally, to describe the Li/Ni cation mixing degree, the I(003)/I(104) peak intensity ratios were analyzed. The I(003)/I(104) peak intensity ratios were used to describe the degree of Li/Ni cation mixing. A higher ratio indicates a smaller mixing degree. Specifically, the ratio in NCA material is 1.31 as opposed to 1.29 for NC material, suggesting that NCA samples possess low-content Li/Ni mixing, which can be attributed to the dispersal of Al^3+^ ions into the bulk phase. This proves that Al- doping not only preserves the original structure of the Ni-rich cathodes but also inhibits cation mixing. It has also been successfully demonstrated that the one-step solid-phase method can synthesize well-ordered structured NCA cathode materials. After 200 cycles, the XRD tests on the cathode materials NC and NCA were conducted, as shown in Appendix A. The peak intensity of both samples decreases, with the NC material exhibiting a weaker intensity, indicating more severe damage to the layered structure after 200 cycles. In addition, (006)/(012) and (018)/(110) of the NCA material still maintain the original cleavage peaks after cycling, while only a few main peaks exist in the NC material. It can be concluded that the NCA manufactured by a one-step solid-state method possesses a stable structure during cycling.

To further observe the homogeneity of the material, scanning electron microscopy (SEM) tests were performed on the 7-Al_2_O_3_-700 °C samples. As shown in Appendix A–g, the samples are composed of sphere-like micrometer-sized particles with a uniform, sphere-like shape and a diameter of approximately 10 µm. Primary particles with an average diameter of roughly 400 nm are responsible for the creation of these particles. Additionally, the elemental distribution on the surface of NCA was examined via EDS analysis (Appendix A–f) to further verify the homogeneity of the as-prepared samples. The EDS images confirm that the Ni, Co, and Al elements are evenly dispersed throughout the NCA surface. Furthermore, the elemental concentration varies slightly from the target content, demonstrating that Al is effectively doped into the bulk phase instead of attaching to the surface.

### 3.3. Enhanced Electrochemical Performance via the One-Step Solid-State Method

To validate the success of the one-step solid-state method in synthesizing NCA materials with excellent electrochemical performance, constant–current charge and discharge tests were conducted on the as-prepared samples. Figure 5a displays the initial discharge-specific capacity for both NC and NCA samples. At a rate of 0.1 C, the initial discharge capacities are 210.9 mAh/g and 204.5 mAh/g for NC and NCA cathodes, respectively. The initial coulombic efficiencies are 87.5% and 86.1%, respectively. The difference in the first discharge capacity may result from the substitution of Al for Co, where Al ions do not engage in the redox reaction during charging and discharging, thus failing to provide capacity. Figure 5b exhibits the dQ/dV curves of the initial cycle that reveal three pairs of redox peaks in the 2.8–4.3 V range, signifying different phase transition processes. The oxidation peak at 3.83 V represents the hexagonal to monoclinic phase transition (H1–M), and the oxidation peaks at 4.01 and 4.20 V represent the monoclinic to hexagonal phase transition (M–H2) and hexagonal to hexagonal phase transition (H2–H3), respectively. The sudden lattice decreases along the c-axis direction during charging. The irreversible H2–H3 phase transition of Ni-rich cathodes may generate and propagate microcracks that can ultimately grow, reach the particle surface during cycling, trigger structural collapse, increase impedance, and aggravate structural instability. However, as shown in Figure 5b, except for the three-phase transition peaks, the irreversible phase transition also occurs around 3.9 V for NC cathodes but not for NCA cathodes. This improbable transfer may result from Al-doping, which has a critical impact on suppressing this irreversible phase transition.

To further reveal the influence of Al, the cells performed 100 cycles at a 1 C rate, as shown in Figure 5c. The NCA cathode with a capacity of 172 mAh/g capacity possesses a high-capacity retention, reaching 97.2% after 100 cycles, and exhibits apparent improvement in cycling stability by doping, while NC cathode decay decreased from a 191 mAh/g to 169 mAh/g with an 88.5% capacity retention. These things considered, the capacity retention in this work outperforms numerous reported studies (Appendix A). Al-doping reduces the initial discharge capacity while substantially improving capacity retention, indicating a considerable impact on cycling stabilities in NCA materials. To highlight the stability of the structure offered by Al-doping, we tested for the rate performance at 2.8–4.3 V, as shown in Figure 5d. As the charge and discharge rates increased (from 0.1 C to 10 C), the discharge capacities of NC and NCA decreased. However, as the rate increases, the capacity difference between the two materials shrinks. More crucially, at a rate of 0.1 C, NC demonstrates significant capacity reductions, while NCA remains stable. This indicates that, after cycling, the NCA material can maintain a more stable structure, elucidating the critical role of Al in the NCA material, including suppressing irreversible phase changes and improving cycling stability. This could imply that Al homogeneously dopes into the NCA cathode manufactured through the one-step solid-state method.

### 3.4. Phase Transition and Impedance Stabilization

XPS tests on NCA and NC cathodes were carried out after 200 cycles to further highlight the impact of Al in the NCA materials manufactured through the one-step solid-state method. The obtained XPS results (shown in Figure 6) indicate that the C–F bond, C–C bond, and C–H bond in PVDF and conductive additive Super P are each assigned to the peaks at 290 eV, 284 eV, and 285 eV, respectively. Additionally, as shown in Figure 6a,d, electrolyte breakdown and carbonate oxidation are related to the C=O, C-O, and CO_3_ peaks at 286 eV, 289 eV, and 290 eV, respectively [30]. The degree of electrolyte breakdown may be determined by the quantities of fluorinated compounds (Li_x_PO_y_F_z_/Li_x_PF_z_/LiF) found on the surface.

In Figure 6b,e, the primary elements of the CEI film at the cathode electrolyte interface, referring to peaks of C–F (678.8 eV), Li_x_PO_y_F_z_/Li_x_PF_y_ (686.5 eV), and Li-F (684.8 eV), are visible in the F 1s spectra. After 200 cycles, the presence of Al in the NCA materials was observed to promote the redox reaction at the interface, as evidenced by the existence of the Li_x_PO_y_F_z_/Li_x_PF_y_ peak at ~687 eV in the NCA, which was not detected in the NC. During the charging and discharging process, Al^3+^ ions do not participate in the redox reaction to provide capacity. However, Al-doping lowers the valence state of Ni, allowing sufficient lithium ions for oxygen redox, which in turn promotes interfacial side reactions. As shown in Figure 6c,f, the O 1s spectra at ~532 eV and ~534 eV for C=O and C-O peaks further confirm that the presence of Al promotes the generation of interfacial side reactions. Therefore, these results suggest that the addition of Al enhances the interfacial stability and improves the cycling performance of NCA materials.

To investigate the Li^+^ transport kinetics behavior of the cathodes after cycling, electrochemical impedance spectroscopy (EIS) was performed on NC and NCA cathodes, as illustrated in Figure 7. Specifically, the tests were executed after 3 cycles at 0.1 C and 200 cycles at 1 C in the voltage range of 2.8–4.3 V. The Nyquist plots after 3 cycles of activation in Figure 7a are made up of two semicircles and a linear straight line. They stand for the resistance of SEI film (R_sf_), charge transfer (R_ct_), and Warburg diffusion (Z_w_). The first semicircle in the high frequency region is related to the surface film resistance of Lithium-ion diffusion due to the solid electrolyte interphase (R_sf_). The second semicircle in the high frequency region is attributed to the charge transfer resistance (R_ct_) that is linked to the kinetics of the electrochemical reaction of both electrodes and variables such as the particle size, phase transition, surface morphology, and the solid–solid connection between active materials. In Lithium-ion batteries, for the first charging, the quantity of Lithium-ion given by the cathode is less than the number of lithium ions travelled back to the cathode after first discharging. This is due to the formation of CEI (solid electrolyte interface). The continuous small-scale reactions then occur between the exposed electrode particles and the electrolyte solution, and the surface impedance is eventually increased by new CEI film generation with cycling. During cycling, CEI formation and growth consume active lithium and electrolyte materials, leading to a fade in capacity, an increase battery resistance, and poor power density. On the electrochemical impedance spectra, it appears that the semicircle representing the surface CEI film resistance (R_sf1_) of the cathode material and the electronic properties of the material (R_sf2_) becomes larger after 200 cycles. The fitting outcomes demonstrate that, as Al is added after activation, the R_ct_ of the electrode reduces. However, after 200 cycles, the Nyquist plots show a combination of three semicircles and a straight line, and the extra semicircles that, after activation, may represent the CEI (lithium-ion passing through the cathode and electrolyte interface) film impedance, as shown in Figure 7b. After 200 cycles, the R_sf2_ impedance of NCA is 131.7 Ω, much lower than that of NC material (192.5 Ω), indicating that the CEI film of NCA material is thinner. This is consistent with the Bode plots (Figure 7c,d) and can be attributed to the frequent generation of CEI films caused by the cracking of Ni-rich materials due to local stress distortion and the sustained small-scale interactions, which gradually increase the surface impedance. Furthermore, the Rct of NCA cathode material increases from 65.2 Ω to 68.2 Ω, while that of NC cathode material increases from 6.8 Ω to 97.0 Ω due to the obstruction of electron transfer resulting from the expansion of the crack during cycling [31]. The detailed data are listed in Appendix A. Therefore, adding Al can effectively stabilize the structure and inhibit irreversible phase transition.

## 4. Conclusions

To reduce the cost of lithium-ion batteries utilizing NCA cathode materials, one possible approach is to simplify the synthesis process. We have developed a one-step solid-state method for synthesizing LiNi_0.9_Co_0.05_Al_0.05_O_2_. We eschew the use of expensive co-precipitation instruments in the commercial co-precipitation method and the pre-sintering step in the traditional solid-phase method. It was determined that Al could be doped into the NC via solid-state calcination by comparing the electrochemical performance and structural characteristics of the as-prepared NC and NCA cathodes. Meanwhile, Al helps suppress the irreversible phase transition, thus improving capacity retention. In addition, the effect of the synthesis conditions on the performance of NCA was extensively investigated in this work. The impact of calcination temperature, ball milling, ball ratios, and aluminum precursors was discussed, and it was found that the calcination temperature had the largest impact on the electrochemical performance. The cathode materials prepared by simple one-step solid-state method had good cycling stability, with a capacity retention rate of 97.2% after 100 cycles at 1 C. The effect of Al_2_O_3_ precursor, 1:7 ball/material ratio, and 725 °C on the electrochemical performances of the synthesized materials are most favorable. This work makes cost reduction in the commercial synthesis of Ni-rich layered oxides accessible and can be used for synthesizing other cathode materials. While this study provides early results for a novel synthesis method, it is important to conduct further research to determine the optimal synthesis conditions in terms of cost, safety, and electrochemical performance.

## Figures and Tables

**Figure 1 materials-16-03079-f001:**
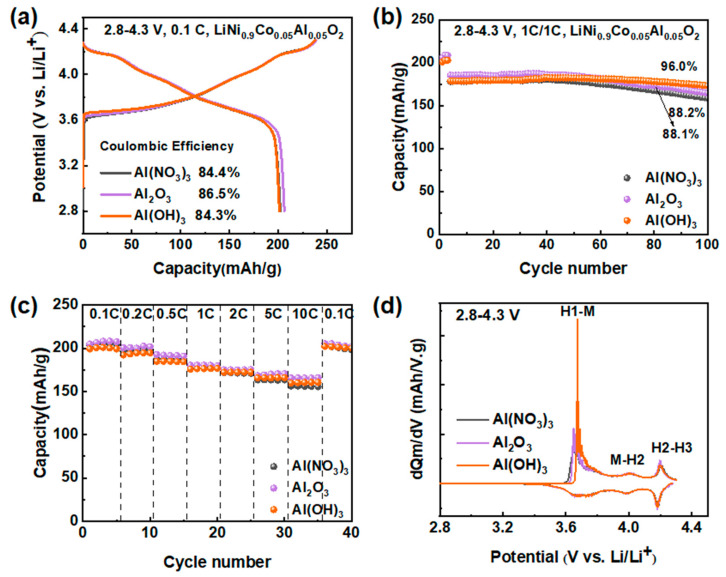
Electrochemical performance of NCA made by three different aluminum precursors Al_2_O_3_, Al(NO_3_)_3_, and Al(OH)_3_ in the 2.8–4.3 V range; (**a**) Charge and discharge voltage profiles of NCA cathodes at 0.1C; (**b**) Cycling performance; (**c**) Rate tests; (**d**) dQ dV^−1^ curves of the NCA cathodes.

**Figure 2 materials-16-03079-f002:**
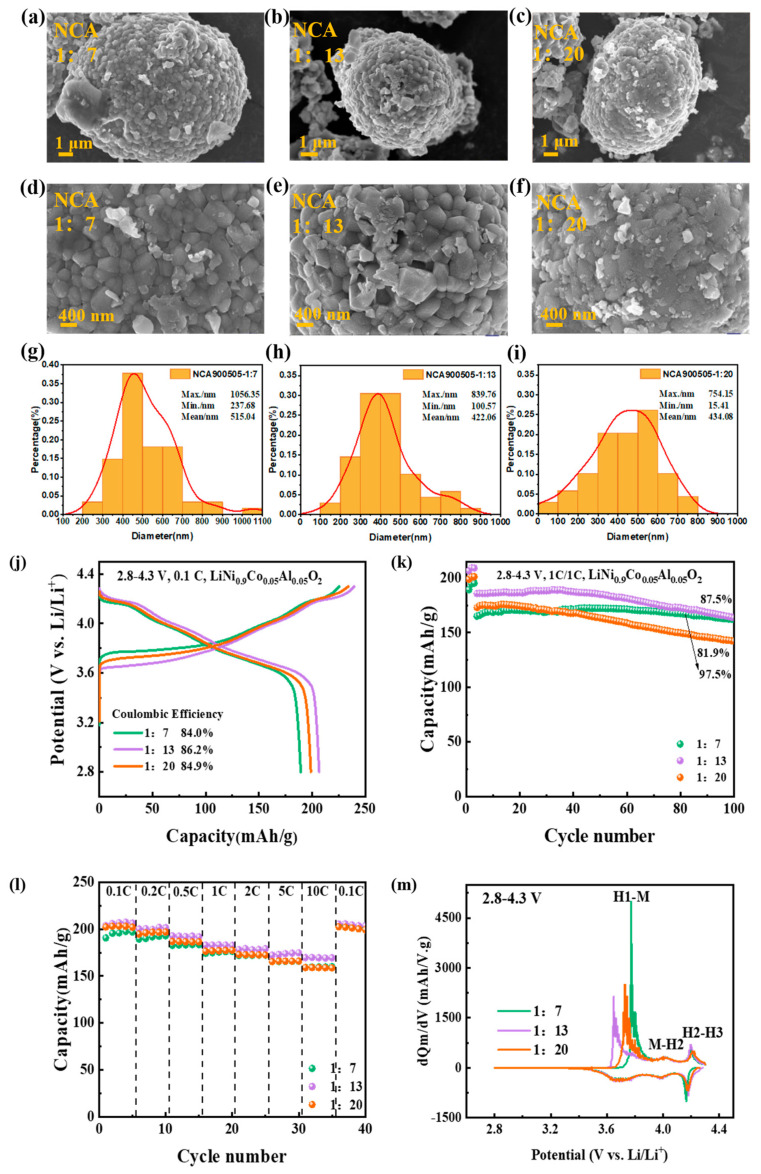
(**a**–**f**) SEM patterns of NCA with different ball/material ratios, and corresponding (**g**–**i**) the average particle size distribution of the primary particles for all the as-prepared samples; Electrochemical performance of NCA made by three different ball material ratios; (**j**) Charge and discharge voltage profiles of NCA electrodes at 0.1 C; (**k**) Cycling performance, (**l**) rate tests, and (**m**) dQ dV^−1^ curves of the NCA electrodes.

**Figure 3 materials-16-03079-f003:**
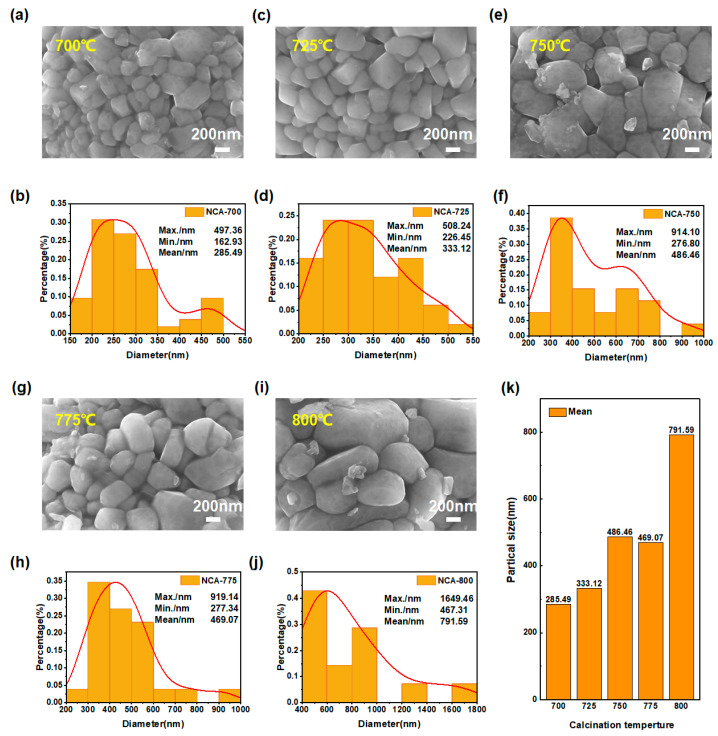
SEM images (**a**,**c**,**e**,**g**,**i**) and particle size distribution (**b**,**d**,**f**,**h**,**j**,**k**) of all as-prepared samples.

**Figure 4 materials-16-03079-f004:**
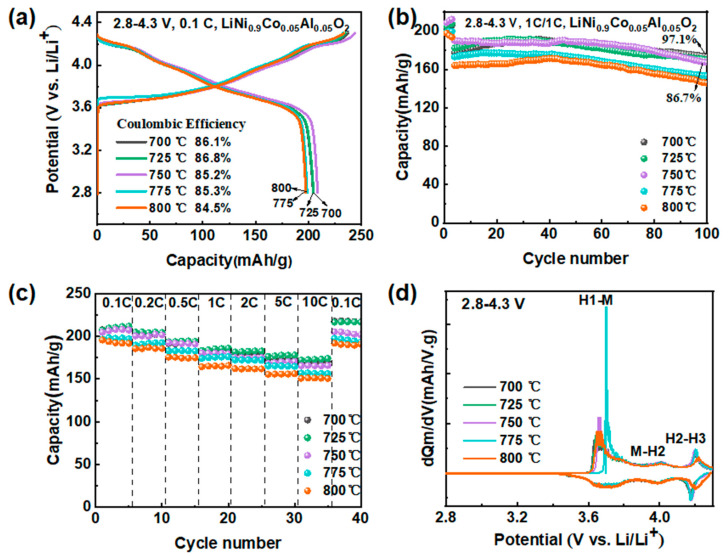
The electrochemical performance of NCA made by five temperatures; (**a**) Charge/discharge voltage profiles of NCA electrodes; (**b**) Cycling performance and (**c**) rate tests of NCA electrodes; (**d**) The dQ dV^−1^ curves of the NCA electrodes.

**Figure 5 materials-16-03079-f005:**
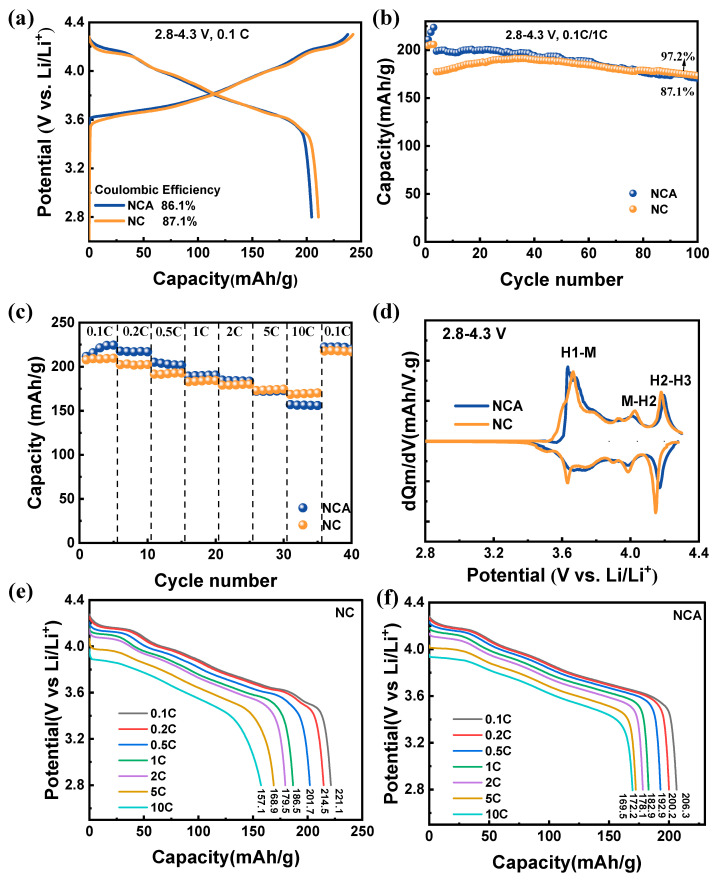
(**a**) Initial charge and discharge curves for NC and NCA cathodes in the range of 2.8–4.3 V; (**b**) The corresponding dQ/dV curves; (**c**) Cycling performance at ambient temperature under 1C; (**d**) Rate performance of NC and NCA cathodes; (**e**,**f**) The specific capacities at different current densities of NC and NCA cathodes.

**Figure 6 materials-16-03079-f006:**
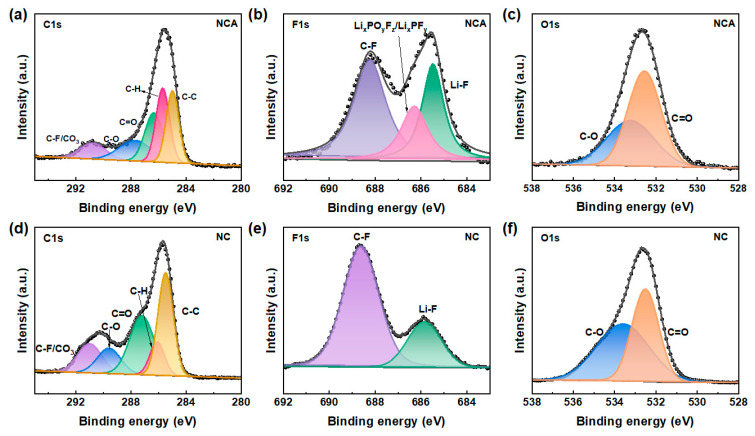
XPS patterns of NC and NCA after 200 cycles; (**a**,**d**) XPS spectra of (**b**,**e**) C 1s, (**c**,**f**) XPS spectra of F 1s, and O 1s.

**Figure 7 materials-16-03079-f007:**
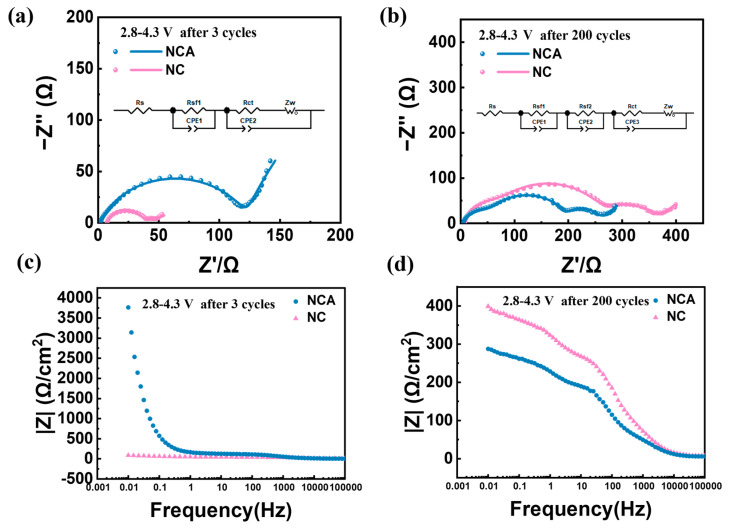
(**a**,**b**) Nyquist plots of NC and NCA electrodes a after activation and b after 200 cycles in the range of 2.8–4.3 V; (**c**,**d**) Bode plots of NC and NCA electrodes c after activation and d after 200 cycles in the range of 2.8–4.3 V.

## Data Availability

Not applicable.

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
