# Peer review of "One-Step Solid-State Synthesis of Ni-Rich Cathode Materials for Lithium-Ion Batteries"

_materials, 2023, doi:10.3390/ma16083079_

Round 1

Reviewer 1 Report

In this work, the authors report one-step solid-state process for synthesizing LiNi0.9Co0.05Al0.05O2 and the impacts of the synthesis conditions on the electrochemical performance for lithium-ion batteries. The manuscript could be accepted for publication in Materials after the following revisions:

Figura 5: no changes in grain sizes are observed. The grain sizes of the three SEM images with different ball material ratios are practically identical.

Why is there no longer cycle data? 100 cycles are relatively low in the current literature, so we can't see the real trend. It is better to provide the electrochemical performance of the electrode for more than 1000 cycles.

Some related works should be compared in the discussion

Please, unify criteria for the reference in the text, for example: line 39 “significance [1, 2].” and line 42: “batteries[3, 4].”

Please, use the subscripts and superscripts correctly in the formulas of the compounds and in the cations, respectively.

“2.2. Electrochemical measurements” section: the font size does not match with of the rest of the subsections. Please modify.

Reviewer 2 Report

The manuscript presents a solid-state process, using one step to synthesize Ni-rich ternary cathode materials. According to the results, the synthesis conditions have substantial impacts on the electrochemical performance. Cyclic stability was verified, with 100 cycles at 1 C. According to the authors, the results show that the one-step solid-state method can synthesize Ni-rich ternary cathode material, with great application potential.

The attached file presents all my doubts, comments, and suggestions.

Round 2

Reviewer 2 Report

Dear editor and authors, after the submitted review, the manuscript must be accepted for publication in the MDPI Journal Materials.